# Algorithm for wireless sensor networks in ginseng field in precision agriculture

**Changcheng Li[1,2,3]\*, Deyun Chen [1]\*, Chengjun Xie[4], You Tang[2,3]**

**1** School of Computer Science and Technology, Harbin University of Science and Technology, Harbin, China, **2** School of Electrical and Information Engineering, Jilin Agricultural Science and Technology University, Jilin, China, **3** Smart Agricultural Engineering Research Center of Jilin Province, Jilin, China, **4** School of computer science and technology, Beihua University, Jilin, China

\* 934853276@qq.com (CL); lichangcheng@jlnku.edu.cn (DC)

**Data Availability Statement:** All relevant data are within the paper.

**Funding:** This work was supported by the National Natural Science Foundation of China (Project No. 60972127, 61072111 and 60672156) and the Key

## Abstract

In the research on energy-efficient networking methods for precision agriculture, a hot topic is the energy issue of sensing nodes for individual wireless sensor networks. The sensing nodes of the wireless sensor network should be enabled to provide better services with limited energy to support wide-range and multi-scenario acquisition and transmission of three-dimensional crop information. Further, the life cycle of the sensing nodes should be maximized under limited energy. The transmission direction and node power consumption are considered, and the forward and high-energy nodes are selected as the preferred cluster heads or data-forwarding nodes. Taking the cropland cultivation of ginseng as the background, we put forward a particle swarm optimization-based networking algorithm for wireless sensor networks with excellent performance. This algorithm can be used for precision agriculture and achieve optimal equipment configuration in a network under limited energy, while ensuring reliable communication in the network. The node scale is configured as 50 to 300 nodes in the range of 500 × 500 m$^2$, and simulated testing is conducted with the LEACH, BCDCP, and ECHERP routing protocols. Compared with the existing LEACH, BCDCP, and ECHERP routing protocols, the proposed networking method can achieve the network lifetime prolongation and mitigate the decreased degree and decreasing trend of the distance between the sensing nodes and center nodes of the sensor network, which results in a longer network life cycle and stronger environment suitability. It is an effective method that improves the sensing node lifetime for a wireless sensor network applied to cropland cultivation of ginseng.

## Introduction

Precision agriculture, a modern agricultural management strategy and operational technology system based on spatial information management and mutation analysis, is of great significance for improving the efficiency of utilization of agricultural resources and ensuring sustainable agricultural development.

In a precision agriculture system, numerous monitoring data need to be collected from the ginseng cultivation fields, including temperature ˚C, humidity %RH, ground temperature ˚C,

Scientific Research Project of Jilin Provincial Department of Education (Project No. JJKH20220390KJ) and Key Project of Jilin Provincial Science and Technology Department (Project No. 20100503) and the Project for Science and Technology Center and Science and Technology Service Platform (Project No. 20180623004TC). The funders had no role in study design, data collection and analysis, decision to publish, or preparation of the manuscript.

**Competing interests:** The authors have declared that no competing interests exist.

ground humidity %RH, illumination Lux, $CO_2$ ppm, wind direction velocity m/s, and precipitation mm. The monitoring environment is diverse and complex. Besides, changes in channel conditions resulting from factors like occlusion are also inevitable during the ginseng growth. Hence, in precision agriculture systems, numerous scholars have focused on improving the energy utilization and network life-cycle in a complex communication context under limited energy conditions of sensor equipment.

In broad terms, precision agriculture refers to an agricultural endeavor that aims to deliver management at an adequate time and location with the appropriate potency. It practices regulated inputs by partitioning the croplands into management blocks or zones rather than offering uniform management. Given the sustained or elevated yield at lower inputs, precision agriculture should mitigate environmental impacts and improve income for the peasants in theory [1]. To achieve precision agriculture, it is necessary to understand the spatial changes in crop and edaphic parameters within croplands [2]. Data communication which serves as an intermediate link between data collection and platform application in precision agriculture, plays a vital role in the stable operation of the system [3]. However, since the sampling points of agricultural information are highly scattered, their geographical distribution is uneven, and power supply is difficult, the construction of wired networks is difficult. Thus, the wireless sensor network has become a basic communication technology in precision agriculture due to its low construction cost, small size, convenient deployment, and flexible network [4, 5]. Wireless Sensor Networks (WSNs), capable of supporting diverse sectors, including manufacturing, education, agriculture, and environmental surveillance, have permanently changed all lives. However, the wireless presence disallows the replacement of the sensor node cells in case the deployment site is remote or unsupervised. To this end, a few studies have attempted to prolong the life expectancy of the nodes. Although cluster-based routing has greatly helped solve this problem, the cluster head (CH) selection can still be bettered through critical variable introduction [6]. However, to improve the flexibility of the network deployment, most devices in the wireless sensor networks use batteries as their main energy source [7]. B. Chatterjee et al. To minimize the network energy expenditure, a co-optimization strategy (real-time) has been formulated, achieving total life-cycle maximization for the batteries. The communication–computation energy tradeoffs for a meshed WSN structure are described by Chatterjee et al. [8], which is arranged throughout a college campus2400-acre in size to monitor the water nitrate content and temperature humidity for smart agricultural purposes via multiple sensors. Deng et al. develop, implement, and elucidate a hardware platform suitable for nodes of a self-powered WSN and mainly aims to prolong the service life of WSN nodes configured in the field context by offering a system for capturing hybrid energy. Apart from minimizing power expenditure through optimization of sensors, microcontrollers, radio frequency (RF) transceivers, etc., the energy sources also need to be taken into consideration, rather than habitual practices of cell replacements or recharges [9]. Exploiting the TENG-based direct sensory transmission (TDST), F. Wen et al. put forward a short-range self-powered (SS) WSN that is free of cells. Wen et al. [10] improved the output of TENG signals and strengthened their frequency through instantaneous short-term discharge by manipulating either a mechanical switch or a diode/switch combo. In this way, signals can be transmitted directly, where no external power supplier or extra wireless module is needed. For energy-harvesting underwater optical WSNs (EH-UOWSNs), N. Saeed et al. develops a localization strategy that is based on received signal strength (RSS), which though encounters great difficulty of predicting ranges due to the optical noise sources in optical devices and the seawater-induced disruption of channels. For viable nodes modeled by utilizing the features of optical communication channels (underwater), their RSSs are determined to achieve the localization of network [11]. In addition to satisfying the basic collection tasks, the device uses a large part of its energy for

transmitting information to other devices. Therefore, in the agricultural environment, the topics of improving the energy utilization efficiency of the sensor devices in the communication process and improving the network life cycle have been rigorously researched by many scholars. In view of the complex and changeable wireless environment in the precision agriculture systems, this paper proposes a novel energy-efficient networking algorithm by dynamically adjusting the network architecture and the signal transmitting power of each node based on the particle swarm optimization (PSO) algorithm, which fully considers the residual energy variation of nodes during communication. Compared to other algorithms, the present method extends the WSN life-cycle greatly.

The remainder of this work is arranged as follows: Section II introduces a precision agriculture system with ginseng cultivation as an example. Section III describes the currently available networking algorithms for the nodes of wireless sensor networks in precision agriculture systems. Section IV describes the node model of the wireless sensor network and introduces the networking algorithm corresponding to the energy-saving wireless sensor nodes based on power control. Results of the validation performed on the algorithm have been presented in Section V by comparing them with the corresponding results obtained using the existing protocols. Section VI provides a summary of this study.

## The precision agriculture system for ginseng

As a crucial ancient Chinese herbal material, Panax ginseng exerts prominent functions on a variety of diseases [12]. As an important Chinese medicinal material, ginseng has the functions of recovering meridians and astringing consciousness loss, replenishing the spleen and kidney, and calming the spirit with marked effects for relieving nerves and improving the immune system [13].

The main ginseng-producing areas are distributed in Jilin, Liaoning, and Heilongjiang in China and South Korea, and North Korea. As a perennial herb, Panax ginseng prefers a cold and sunny climate, and ginseng growth requires strict illumination conditions. In addition, in terms of its growth environment, the relative water content of the soil needs to be above 70–80%, and the pH value required is preferably 6.0–6.5. In order to ensure the sustainability of ginseng cultivation, China has undergone a major transformation from "cutting forest to grow ginseng" to "cultivating ginseng in crop fields". Ensuring that the trace elements are available in sufficient quantities in the soil and the growth environment in the farmland meets the requirements for ginseng cultivation are the key aspects that impact the emergence rate and yield of ginseng.

As the Internet of Things (IoT) and novel sensing technologies are evolving persistently, a good solution has been provided for ginseng cultivation in crop fields. As shown in Fig 1, the precision agriculture system consists of a sensing layer, a transport layer, as well as a platform for data application. The sensing layer consists of a series of sensing devices and mainly implements the collection of key information such as the soil moisture, its pH value, and the light intensity during growing ginseng in the crop fields. The transport layer is the main transmission path for connecting the collected data to the application platform with data forwarding and intermediate data processing functions. The application platform reprocesses the data according to actual needs and provides a series of targeted and professional functions of data application such as environmental monitoring systems and expert systems in agriculture.

In the precision agriculture system for ginseng cultivation in a farm field, the influence of the cristate leaf on the transmission of wireless signals should not be ignored during its growing process due to the diversity of the collection parameters such as light, soil moisture, and pH value, and the deployment of sensors at different heights. Furthermore, the limited energy

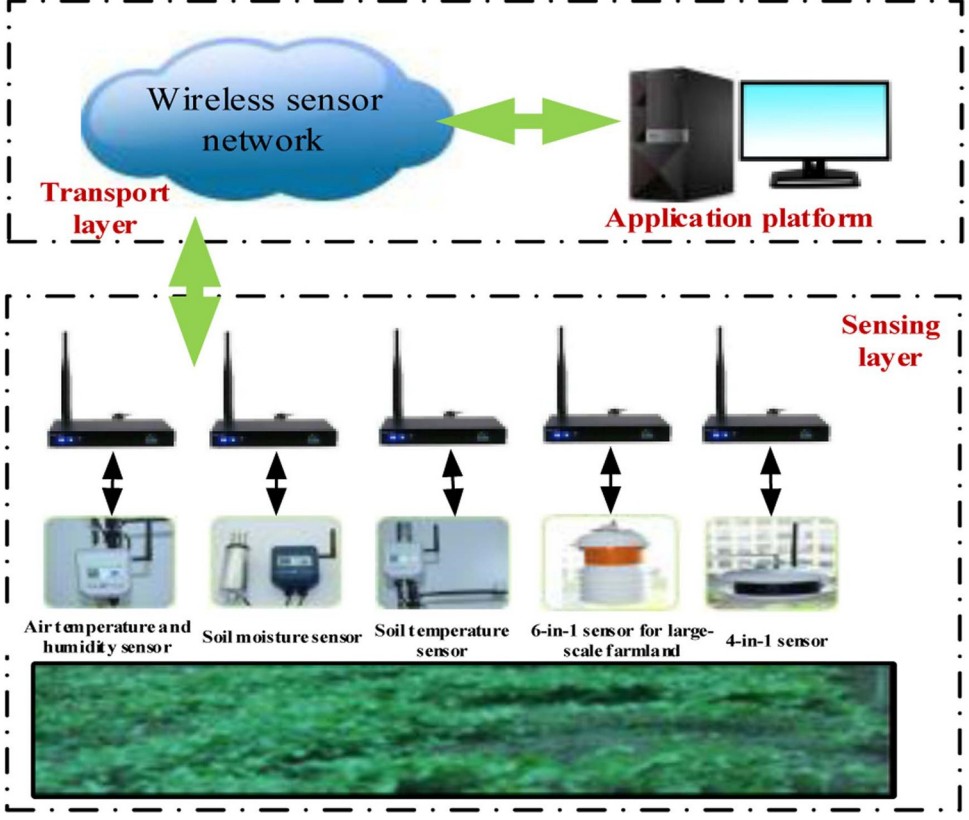

**Fig 1. Structure of a precision agriculture system.**

available to the devices in the wireless sensor network demands a set of efficient transmission strategies. In order to ensure the long-term durability of the devices under different pressures, extensive scholarly efforts have been devoted to the energy conservation in these devices from the aspects of hardware design, network routing, and data with compressive sensing. This study presents an energy-saving networking algorithm for a precision agriculture system that combines the characteristics of the wireless sensor devices and the advantages of the particle swarm optimization technique.

## Related works

In the research studies conducted on the routing algorithms of wireless sensor networks based on precision agriculture systems, based on the concept of a mobile node, Prasad R et al. used miniature aerial vehicles as the receiver of regional data [14]. This reduces the energy consumption caused by multi-hop data transmission and ensures reliable network communication with low power consumption. However, periodic small-scale collection systems, such as ginseng growing, undoubtedly increase the equipment cost. In order to achieve balanced energy consumption by the network and to avoid the "premature death" of nodes caused by the "hot spot" problems, a scheme called energy-efficient load-balanced clustering (EELBC) has been formulated, which upgrades the network energy efficiency through number equilibration of cluster members [15]. Considering the shortcomings of the low-power adaptive clustering hierarchy (LEACH) protocol in the random cluster head mechanism.

Kandris et al. have modeled the network as a linear system [16] and, starting from the current and future residual energy of the nodes, have proposed the equalized cluster head election

routing protocol (ECHERP) that effectively improves the overall life cycle of the network. A differential evolution (DE)-based clustering strategy has been proposed [17] for WSNs, which inhibits the rapid channel (high-load) death to extend the network life-cycle. However, the wireless presence disallows the replacement of the sensor node cells in case the deployment site is remote or unsupervised. To this end, a few studies have attempted to prolong the life expectancy of the nodes. Although cluster-based routing has greatly helped solve this problem, the cluster head (CH) selection can still be upgraded through the critical variable introduction. Moreover, the main emphasis has been laid on the choice of CH or the inter-node transmission of data. For network efficiency optimization, the potential of meta-heuristic strategies can be bright [18]. Furthermore, the layout of sensors has been explored under restricted conditions of sensor and App communication ranges [19]. In Double-tiered WSNs, the algorithms are superior to the established clustering approaches, including Agglomerative Clustering, Minimum Energy Routing, PSO, Divisive Clustering, as well as Relay Node Placement and its Improved approach regarding mean performance [20]. On surface networks, the paths of varying homotopy types are calculated by developing an approach that is free of GPS and tessellation. According to the virtual planar coordinate calculations of nodes, a data package is sent by a source node to its destination in a greedy manner. Upon failure of an existing path, other greedy path options (varying homotopy types) are available for the source node to achieve the package delivery. The proposed distributed algorithms are scalable to the size of surface networks and their genus number. With the proposed design, a novel scheme of multichain routing can be generated for data package transmission in the energy-balanced WSNs. The multichain routing technique can compute the optimal inter-node distance of transmission through mathematical modeling. Regarding the design standard for optimal distance, the farthest sending node within a communication scope and the direction to sink are chosen so that the multichain architectures can be established with precise direction, distance, and multipath. Shorter chain routing allows cutting unwarranted energy consumption, thereby achieving life-cycle extension for the global network [21]. Initially, a framework for smart agriculture is designed based on SWIPT. The next step involves the exploration of an energy performance-optimizing strategy for green communication. In this process, the pairing of subcarriers is optimized together with their power allocation. A communication procedure comprises two stages. To tackle the optimization problem proposed, a Lagrangian dual formulation is utilized to put forward an efficient algorithm of iterative optimization [22].

A composite routing measurement method has been proposed for combining energy sensing, reliability awareness, robustness awareness, and resource awareness, which increases the adaptability of the system to different scenarios [23]. Azharuddin et al. [24] adopted distributed clustering and routing ideas from the perspective of fault tolerance to solve the problem of network reorganization after the failure of cluster head nodes. Under a proposed model, a brand-new problem of sensor clustering is formulated, where the sensors are divided into varying clusters, and a sensor enabling transmission quantity maximization of redundant suppressed data is chosen for each cluster [25]. Li et al. [26] develop an effective routing scheme called DMARL for UOWSNs, which exploits multiagent reinforcement learning. Initially, the network is created as a distributed multiagent architecture through modeling. The design of the routing scheme takes into account the link quality and residual energy to better accommodate the dynamic environment and help better extend the network lifespan. For faster convergence of the algorithm for reinforcement learning, two additional optimization measures are developed as well. For the distributed system, they offer a reward mechanism on this basis. Cao et al. [27] put forward enhanced versions of distributed parallel algorithms after checking the prior knowledge-based and the stochastic grouping and an additional optimizer. Their algorithms are superior to counterparts in terms of performance, which exhibit remarkably

shorter time expenditure than the serial algorithms owing to the distributed parallelism nature. Hence, within a considerably restricted time, a higher performance can be attained by their algorithms. An optimization problem is formulated by Baek et al. [28] to maximize the post-data transmission minimum residual sensor energy in UAV routing (energy-saving scheme), which depends on data collection and is constrained by the travel distance of UAV. Focusing on the life-cycle maximization of a WSN, Zhang et al. [29] develop a sensor configuration scheme called EDAGD, which comprises several algorithms. Its performance is superior to the classic random deployment algorithms. Künzel et al. [30] propose a Q-learning reliable routing strategy, which has a weighting agent for achieving weight adjustment of a most-advanced graph-routing algorithm. The sets of weights are denoted by the agent state, and in the process of network operation, the weights are altered by actions. Upon shortening of mean network latency or extension of a network life expectancy, the agent is rewarded accordingly. Liu et al. [31] develop TPE-FTED, a fault-tolerant algorithm for event region detection, where the faulty nodes are recognized based on the extraction of trajectory patterns. By learning the probabilistic model online, the sensing value distributions in varying sensing states are elucidated by individual nodes. A trajectory can be generated with a particular group of probabilistic models, which is indicative of the occurrence of a special event. TPE-FTED performs the pattern matching and examines the spatiotemporal restraints by utilizing the implicit information from the trajectories, so that the faulty node declaration can be recognized. For efficiency enhancement of network having QoS parameters (multi-constrained), the problem is modeled as a multi constrained optimal path issue and preserved by developing an algorithm based on distributed learning automaton (DLA). Exploiting the pros of DLA, the proposed method finds the minimum number of nodes to conform to the QoS requirements. A few QoS routing constraints are considered during path selection, such as end-to-end delay and reliability. Algorithm outperforms existing advanced algorithms regarding energy efficiency and end-to-end delay [32]. Given above, the use of UAVs for data acquisition in dense WSNs is proposed, where the compressive data gathering (CDG)based on projections serves as a new solution technique. Data en-route are gathered from a big sensor node set by CDG to choose the projection nodes working as cluster heads (CHs), so that the number of required transmissions can be lowered to remarkably save energy and prolong the network life-cycle [33]. By fusing analysis of edge data with that of cloud data, a brand-new method for automated detection of anomalies in heterogeneous WSNs is developed. The edge data analysis exploits an entirely unsupervised algorithm of artificial neural network, while with the cloud data analysis, the multi-parameterized edit distance algorithm is utilized [34]. However, none of the abovementioned algorithms take the transmit power tunability of wireless sensor devices into consideration, resulting in insufficient utilization of the device performance during network construction. In this study, taking the characteristics of power control and wireless sensor networks into consideration, the network has been established from three aspects, namely, device performance, energy consumption, and location information. When wireless communication is inadequate, it is necessary to increase the transmission power of the high-energy devices to reduce the unnecessary data forwarding nodes. On the other hand, when wireless communication is adequate, it is necessary to decrease the transmission power of the neighboring devices for reducing the overall increased energy consumption in the network caused by the data loss or retransmission resulting from the interference of the adjacent communication channels.

## The proposed algorithm

This section describes the algorithm proposed herein for an energy-efficient WSN that is applied to ginseng cultivation in the precision agriculture system. The algorithm incorporates

power control into the network construction by adjusting the communication radius of each sensor node to achieve balanced energy consumption in the network. In addition, the proposed algorithm preferentially selects the forward and high-energy nodes as the cluster heads or the data forwarding nodes, so that the transmission orientation and energy expenditure of the nodes can be taken into account.

## Abstraction of the network model

Fig 2 shows a schematic of the network structure of the precision agriculture system based on the clustering model. Devices in the network can be divided into three types: central node, cluster head node, and cluster member node. The central nodes, having a fixed power supply and strong information processing capabilities, provide data interaction between the wireless sensor network and the internet. The same hardware structure is employed for the cluster head nodes and the cluster member nodes in the network with the battery power statically distributed in a certain area. The cluster member node is mainly responsible for responding to the collection instruction and sending the response information to the corresponding cluster head node. In addition to the abovementioned functions, the cluster head node is also responsible for the functions of member management and data forwarding within the cluster.

## Energy-based model for the network

For energy expenditure analysis of the wireless sensor nodes in the WSN, the widely acknowledged energy expenditure model for wireless communication is adopted in this study [16].

$$\begin{cases} E_{tx}(l, d) = \begin{cases} lE_{elec} + l\varepsilon_{fs}d^2 & d < d_c \\ lE_{elec} + l\varepsilon_{two-ray}d^4 & d < d_c \end{cases} \\ E_{rx}(l) = lE_{elec} \end{cases} \quad (1)$$

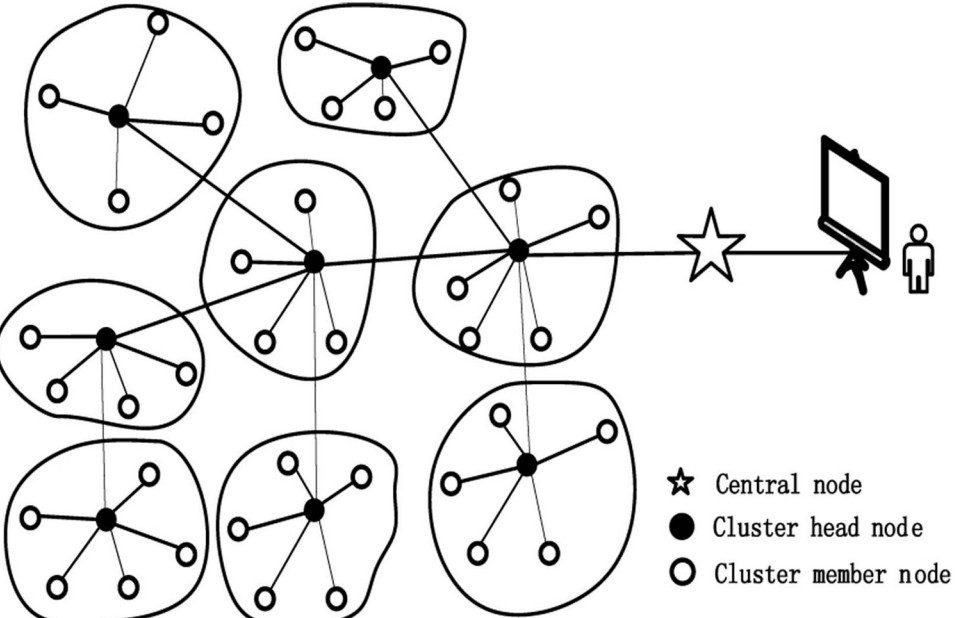

**Fig 2. Network diagram of the precision agriculture system.**

Where $E_{tx}$ and $E_{rx}$ represent the energy expenditures from transmission and reception of data, respectively, $l$ is the length of the transmitted data, $E_{elec}$ denotes the energy expenditure by the node for each bit of forwarded/received data, and $d$ denotes the distance between sending and receiving nodes. In case the distance is below the threshold $d_c$, we use a free space model; otherwise, we use a multipath model. $\varepsilon_{fs}$ and $\varepsilon_{two\text{-}ray}$ are the power magnifications in the different models. $d_c$ can be calculated using the following formula:

$$d_c = \sqrt{\frac{\varepsilon_{fs}}{\varepsilon_{two-ray}}} \tag{2}$$

Because $E_{elec}$ depends on the hardware, such as a digital coder, modulator, and filter, the default value of this system is fixed in this study.

## Description of the proposed algorithm

In order to facilitate research and analysis in this study, a wireless sensor network has been abstracted with a two-way communication function as an undirected connectivity graph, $G = (V, E)$, where $V$ represents a set of all wireless sensor devices, and $E$ is a possible set of communication paths between all devices. The transmit power can be altered for all wireless sensor devices by changing the communication radius. For a unified analysis of energy consumption in the network, the network life cycle is defined as the ratio of the number of transfers from the commencement of data transmission to the "death" of the first network node.

In order to quantitatively represent the transmit power of the device, the "power adjustment factor" for each wireless sensor device is set according to the following expression:

$$w = \frac{Device's\ current\ transmit\ power}{Reference\ transmit\ power} \tag{3}$$

And its value is in the range of [0.5, 1.5]. With the introduction of the "power adjustment coefficient", the wireless network construction becomes a non-deterministic polynomial problem. In order to achieve rapid optimization of the network structure, particle swarm optimization has been introduced as the main solution algorithm. Fig 3 illustrates the flowchart of our algorithm.

**Phase I: Initialization of the proposed algorithm.** Suppose the number of particles in the PSO algorithm is $m$, the maximum number of cycles is $p$, and the network optimization index is $f = \max(Life)$, where $Life$ is the value of the life cycle of the network. The position update formula of each particle is given by

$$W^{k+1} = W^k\lambda + C_1(W_p - W^k) + C_2(W_g - W^k) \tag{4}$$

Where $W$ is the "power adjustment factor" matrix, $W = \{w_i \,|\, i = 1,\ldots,N\}$, $N$ is the number of nodes in the sensor network, $\lambda$ is the inertia coefficient, $C_1$, and $C_2$ are the self-learning factors, and $W_p$ and $W_g$ represent the individual optimal value and the group optimal value, respectively.

**Phase II: Network construction under particle condition.** (1) Cluster head selection phase

Step 1: The central node sends a request for route creation to the nodes within the communication range in a broadcast form, and each node obtains the information of the minimum hops, $h_{min}$, between the central node and itself.

Step 2: The communication radius, $R_i$, of each node is calculated according to the "power adjustment factor" matrix, and the degree of discretization, DisRatio, among a node and its

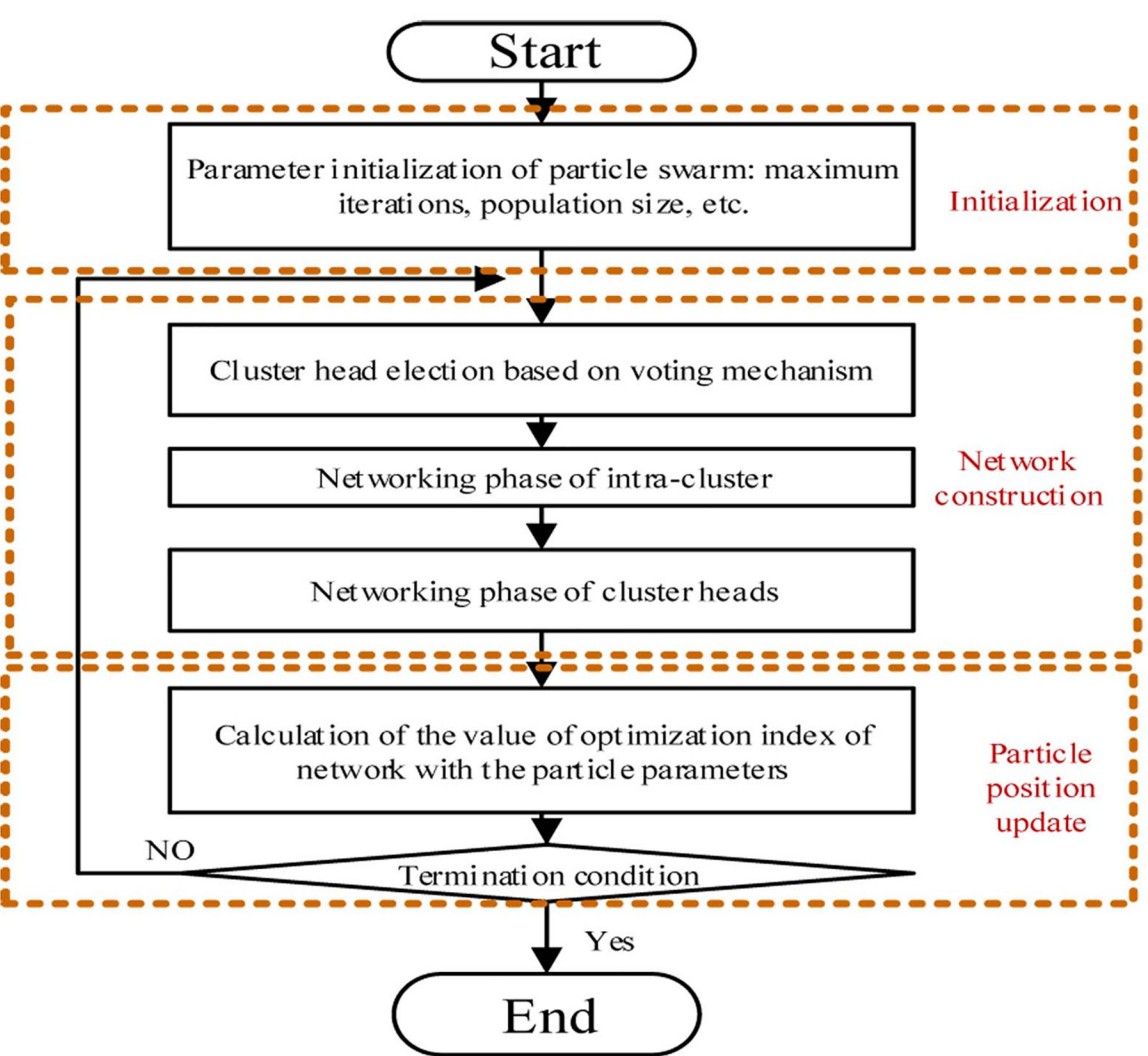

**Fig 3. Flowchart of the proposed algorithm.**

adjacent nodes is obtained as follows:

$$DisRatio(i) = \frac{\sum\limits_{j=1, j \in Neighbor(i)}^{N} dis(j, i)}{N \bullet R_i} \tag{5}$$

where dis*(j, CH$_i$)*denotes the communication distance between node *i* and its adjacent node *j*, and *N* denotes the number of neighboring nodes. The smaller the index, the higher is the concentration of the neighboring nodes and the cluster heads.

Step 3: Within the current communication radius, the maximum number of data transfers, *Count* (*i*), is estimated considering the node *i* as the cluster head node:

$$Count(i) = \frac{E(i)}{(N-1)(E_{rx} + \eta E_{tx})} \tag{6}$$

Where *E(i)* refers to the residual energy of the current node, *N* denotes the number of

adjacent nodes, $\eta$ is the data fusion coefficient, and $R_i$ is the estimated communication distance when data is sent.

Step 4: The "fitness factor", $fitness(i)$, is calculated considering the node $i$ as the cluster head:

$$fitness(i) = DisRatio(i)^{\alpha} Count(i)^{\beta} \tag{7}$$

where $\alpha$ and $\beta$ are the proportional coefficients of each item, and the system assumes their values to be –1 and 1, respectively.

Step 5: Among the neighboring nodes, the one with the largest "fitness factor" is voted as the cluster head node by all the nodes in the network.
(2) Networking phase

Step 6: The selection coefficient, $C(j, i)$, of each cluster head node from the neighboring nodes is calculated using the cluster member node $j$, and the cluster head node with the larger coefficient is selected for networking.

$$C(j, i)_{choose} = \frac{E(i)^{\varphi}}{E_{tx}(j, i)} \cdot \left(\frac{h(j)_{min}}{h(i)_{min}}\right)^{\gamma} \tag{8}$$

where $E_0(i)$ refers to the residual energy of cluster head node $i$, $E_{tx}(j,i)$ represents the communication loss between the nodes, and $h(j)_{min}/h(i)_{min}$ guarantees that the cluster head, which is closer to the central node is selected for networking. $\varphi$ and $\gamma$ are the proportional coefficients of each item, and the system assumes their values to be –1 and 1, respectively.

Step 7: The aggregate information of the adjacent cluster heads obtained from the cluster head node (with $m$ as an example) is used for calculating the selection coefficient $CH(m, n)$ of the candidate forwarding node, and the cluster head node with maximum coefficient is chosen as the data forwarding node.

$$CH(m, n) = count(n) \cdot \left(\frac{h(m)_{min}}{h(n)_{min}}\right)^{\theta} \tag{9}$$

where $count(n)$ is the maximum number of data transmissions that can be performed after the node $n$ is selected as the cluster head, $h(m)_{min}/h(n)_{min}$ guarantees that the cluster head, which is closer to the central node, is preferentially selected to forward data, $\theta$ is the proportionality factor for this item, and its value has been taken to be 1 in this study.

**Phase III: Particle update phase.**

Step 1: From the abovementioned procedure, the life cycle values of each network are calculated under the condition of $m$ different particles, along with a record of the individual optimal value, $W_p$, and the group optimal value, $W_g$, in the objective function as well as their corresponding "power adjustment factor" matrices, with the number of cycles increased by one.

Step 2: The particle position is updated using the particle update formula, and the steps in Phase II are repeated until the program termination conditions are met.

In the clustered network structure, the functional characteristics of the cluster head node determine whether its energy consumption is much larger than the cluster member nodes. In this study, the cluster head rotation mechanism has also been used for optimizing the network. The rotation threshold of the cluster head is set to $q$. When 1/5 of the network life cycle is

greater than $q$, either 1/5 of the life cycle of the nodes in the network or $q$ instances of data transmissions are used as the trigger conditions for the network reconstruction.

## Performance analysis of the algorithm

### Description of the experimental parameters

For verification analysis of our algorithm, the MATLAB is used for conducting the simulations in the range of $500 \times 500$ m$^2$ and a comparison has been made between the results obtained using the proposed algorithm and the typical LEACH [35], BCDCP [36], and ECHERP [37] algorithms. During the simulations, each wireless sensor node in the WSN is identical regarding initial energy and performance of wireless communication and is randomly fixed in the entire area. The physical structure of the precision agriculture system network is as follows: The central node is located outside the measurement area equipped with an independent power supply. The physical position of wireless sensor equipment in the WSN cannot be changed, but it has the function of transmitting power adjustment using multi-hop as the method of transmission. The parameter settings in the experiment are shown in Table 1.

As shown in Table 1, 50 to 300 wireless sensors are deployed in the network at random; and their physical position cannot be changed, but such equipment has the function of transmitting power adjustment; and power consumption of the node for transmitting/receiving 1bit data $E_{\text{elec}} = 50$nJ/bit.

In the proposed algorithm, Particle Swarm Optimization (PSO) is taken as the basis of algorithm optimization; and in the particle update rules, the number of particles $m = 10$, selected self-learning factors $C_1$ and $C_{2} = 0.5$, inertial coefficient $\lambda = 0.5$, and the maximum number of optimizations of the particle $p = 30$.

### Performance analysis of the networking algorithm

**Effect of the network size.** The number of network nodes, which is an important factor that affects the communication distance between the network nodes, has an important effect on the network life cycle. For networking performance verification of our algorithm over varying sizes of the network, the LEACH, BCDCP, and ECHERP algorithms were used in the simulations with 50–300 nodes selected in an area of $500 \times 500$ m$^2$. The simulation results are shown in Fig 4. With the gradual addition of the network nodes, the network life cycle for different algorithms is observed to increase gradually, and the proposed algorithm exhibits superior performance in improving the network life cycle. The algorithm comparison on network life cycle under different numbers of nodes is shown in Table 2.

**Effect of the position of the central node.** In the precision agriculture system, the location of the host determines whether the central node should be deployed outside the monitoring range of the wireless sensor network or not. As an important part of the sensor network, the distance of communication between the central node and the sensing equipment greatly

**Table 1. Parameter settings.**

| Parameter | Value |
|---|---|
| No. of nodes | 50–300 |
| $C_1, C_2$ | 0.5 |
| $m$ | 10 |
| $p$ | 30 |
| $E_{elec}$ | 50 nJ/bit |
| $\lambda$ | 0.5 |

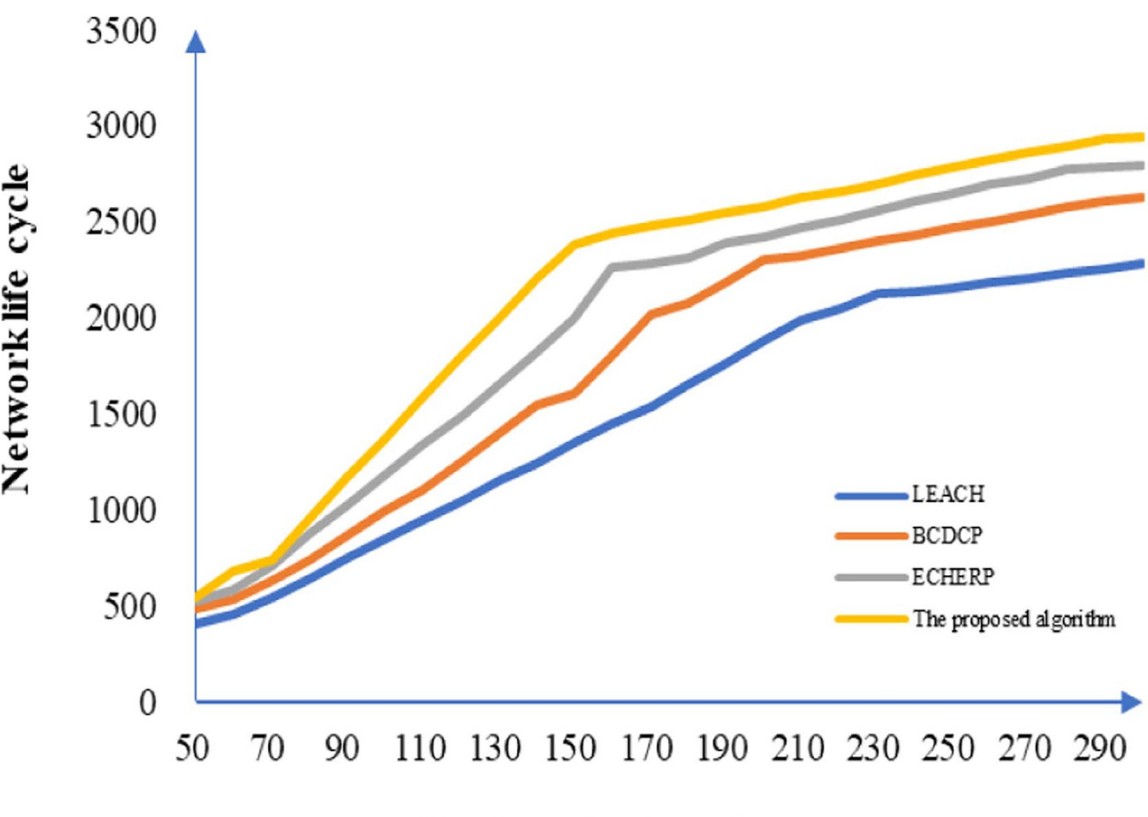

**Fig 4. Network life cycle as a function of the number of nodes deployed.**

affects the size of the network life cycle. To confirm how this factor affects the network life cycle, a distance of 50 ~ 170 m from the edge of the monitoring range was selected for the simulation and the results thus obtained are shown in Fig 5. As the distance between the device and the position of the central node increases gradually, the network life cycle is observed to decrease gradually. When the position is greater than 130 m, the downward trend becomes steeper. The role of the distance between sensing and central nodes appears crucial. The algorithm proposed in this work exhibits a lower decline rate and downward trend than the compared algorithms. The algorithm comparison on network life cycle under different distances of the central node is shown in Table 3.

**Analysis of transmission frequency of the network node.** The analysis of the data transmission process of all nodes in the network requires verification of the balance in the energy consumption during data transmission by the network. In this study, 200 sensor nodes were

**Table 2. Algorithm comparison on network lifecycle under different numbers of nodes.**

| Number of nodes / Algorithm | 50 | 100 | 150 | 200 | 250 | 300 |
|---|---|---|---|---|---|---|
| LEACH | 471 | 830 | 1380 | 1890 | 2198 | 2258 |
| BCDCP | 488 | 960 | 1603 | 2260 | 2460 | 2680 |
| ECHERP | 493 | 1180 | 1900 | 2301 | 2597 | 2732 |
| Proposed algorithm | 502 | 1302 | 2301 | 2587 | 2730 | 2886 |

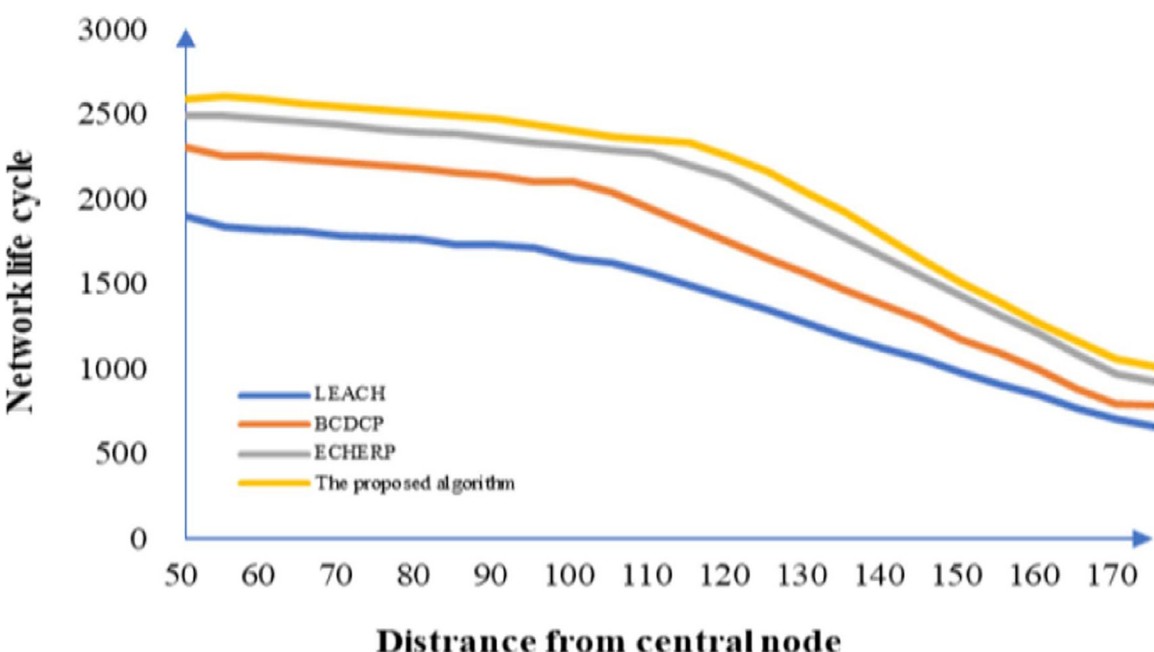

**Fig 5. Network life cycle as a function of the distance of the device from the position of the central node.**

selected in the range of 500 × 500 m$^2$ in the simulation, and the central node was assumed to be 100 m away from the edge of the monitoring range. The failure time of the network node for different algorithms is shown in Fig 6. Compared to the other algorithms, the large-scale failure of the network nodes in the algorithm proposed in this study and the breakdown in its communication occur later with the increasing frequency of data collection. The algorithm comparison on network life cycle under different distances between the central node and other nodes is shown in Table 4.

## Conclusion

In order to enable limited energy usage by the wireless sensors in the precision agriculture system, an algorithm for high-efficiency wireless sensor networks for a ginseng field has been proposed in this study. Exploiting sensor nodes' adjustable transmission power, the particle swarm optimization technique has been used to achieve the communication range adjustment of sensing devices in the network. From the aspect of reliable communication, the optimal deployment of the limited energy of the devices in the network has been achieved. The simulation results show that compared to the existing LEACH, BCDCP, and ECHERP algorithms, the proposed algorithm has a higher life cycle for different network sizes with good performance at different deployment locations of the central node.

**Table 3. Algorithm comparison on network life cycle under different distances of central node.**

| Distance of central node<br>Algorithm | 50 | 75 | 100 | 125 | 150 | 175 |
|---|---|---|---|---|---|---|
| LEACH | 1896 | 1800 | 1687 | 1409 | 1001 | 690 |
| BCDCP | 2386 | 2205 | 2098 | 1689 | 1201 | 834 |
| ECHERP | 2500 | 2402 | 2385 | 2045 | 1480 | 960 |
| Proposed algorithm | 2598 | 2523 | 2443 | 2303 | 1513 | 1000 |

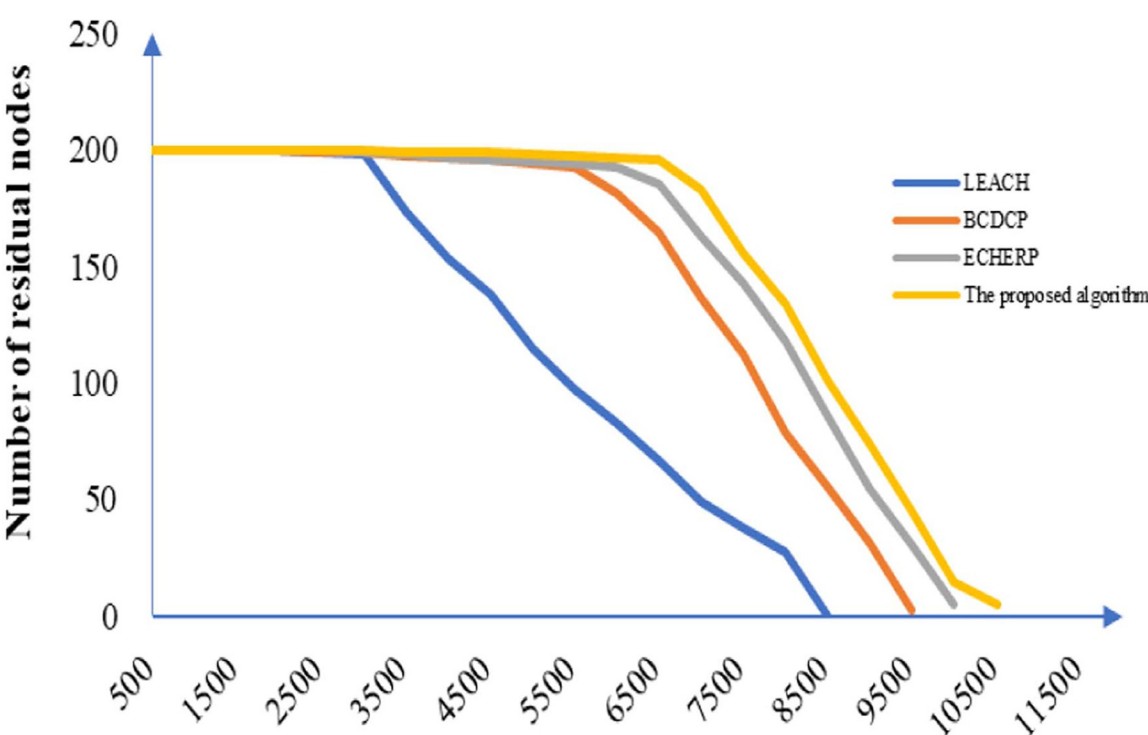

**Fig 6. Failure time of all nodes in the network corresponding to different algorithms.**

**Table 4. Algorithm comparison on network life cycle under difference distances between central node and other nodes.**

| Algorithm \ Number of other nodes | 150 | 100 | 50 | 0 |
|---|---|---|---|---|
| LEACH | 4102 | 5540 | 7054 | 8579 |
| BCDCP | 6940 | 7602 | 8569 | 9509 |
| ECHERP | 7340 | 8402 | 9230 | 10301 |
| Proposed algorithm | 7580 | 8501 | 9474 | 10503 |

Therefore, the environmental growth variables and indices of farm fields in Jilin, Liaoning, and Heilongjiang in China, and the remote mountainous areas in South Korea and North Korea, have been monitored in real-time using the proposed algorithm. The optimal growth environment of the ginseng fields has been improved using the sensing nodes of the wireless sensor network, which has a significant application value in increasing the yield of ginseng and reducing labor costs.

## Author Contributions

**Conceptualization:** Changcheng Li, Deyun Chen.

**Data curation:** Changcheng Li, Deyun Chen, Chengjun Xie.

**Formal analysis:** Changcheng Li, Deyun Chen, You Tang.

**Methodology:** Changcheng Li, Chengjun Xie.

**Writing – original draft:** Changcheng Li, Deyun Chen, Chengjun Xie, You Tang.

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
