## [Decision Letter · Decision Letter 0]

26 Nov 2021

PONE-D-21-28152Algorithm for Wireless Sensor Networks in Ginseng Field in Precision AgriculturePLOS ONE

Dear Dr. Chen,

Thank you for submitting your manuscript to PLOS ONE. After careful consideration, we feel that it has merit but does not fully meet PLOS ONE’s publication criteria as it currently stands. Therefore, we invite you to submit a revised version of the manuscript that addresses the points raised during the review process.

We look forward to receiving your revised manuscript.

Kind regards,

Chi-Tsun Cheng, Ph.D., M.Sc., B.Eng.

Academic Editor

PLOS ONE

Journal Requirements:

2. Thank you for stating the following in the Acknowledgments/ Financial Disclosure Section of your manuscript: 

This work was supported by the National Natural Science Foundation of China (Project No. 60972127, 61072111 and 60672156) and Key Project of Jilin Provincial Science and Technology Department (Project No. 20100503) and the Project for Science and Tech-nology Center and Science and Technology Service Platform (Project No. 20180623004TC). The funders had no role in study design, data collection and analysis, decision to publish, or preparation of the manuscript.

This work was supported by the National Natural Science Foundation of China (Project No. 60972127, 61072111 and 60672156) and Key Project of Jilin Provincial Science and Technology Department (Project No. 20100503) and the Project for Science and Tech-nology Center and Science and Technology Service Platform (Project No. 20180623004TC). The funders had no role in study design, data collection and analysis, decision to publish, or preparation of the manuscript.

4. We note you have included a table to which you do not refer in the text of your manuscript. Please ensure that you refer to Table 1 in your text; if accepted, production will need this reference to link the reader to the Table.

Additional Editor Comments:

Further elaboration on the experiment setup is required. The rationale of the selected evaluation criteria has not be explained clearly. The connection between the results (i.e. figures) and the intrepretations in the analysis section should be highlighted and extended further.

Reviewers' comments:

Reviewer's Responses to Questions

**Comments to the Author**

1. Is the manuscript technically sound, and do the data support the conclusions?

Reviewer #1: Yes

Reviewer #2: Yes

2. Has the statistical analysis been performed appropriately and rigorously? 

Reviewer #1: Yes

Reviewer #2: N/A

3. Have the authors made all data underlying the findings in their manuscript fully available?

Reviewer #1: Yes

Reviewer #2: Yes

4. Is the manuscript presented in an intelligible fashion and written in standard English?

Reviewer #1: Yes

Reviewer #2: Yes

5. Review Comments to the Author

Reviewer #1: The paper proposed an algorithm for high-efficiency wireless sensor networks for a ginseng field explained well. But

parameters settings and description of the experimental parameters need more explanation.

May be accepted for the publication.

Reviewer #2: The paper presents Algorithm for Wireless Sensor Networks in Ginseng Field in Precision Agriculture. The overall architecture and main component are describes. Since statistical analysis should be performed rigorously, you can add a table contains many aspects comparing existing algorithms to the proposed algorithm and specify the method of testing that had been used in the study as well.

6. PLOS authors have the option to publish the peer review history of their article (what does this mean?). If published, this will include your full peer review and any attached files.

Reviewer #1: No

Reviewer #2: No

---

## [Author Response · Author response to Decision Letter 0]

16 Dec 2021

List of Responses

Dear Editors and Reviewers:

Thank you for your letter and for the reviewers’ comments concerning our manuscript entitled “Algorithm for Wireless Sensor Networks in Ginseng Field in Precision Agriculture”, (Manuscript Number: PONE-D-21-28152). Those comments are all valuable and very helpful for revising and improving our paper, as well as the important guiding significance to our researches. We have studied comments carefully and have made corrections which we hope meet with approval. Revised portions are marked in red in the paper. The main corrections in the paper and responses to the reviewer’s comments are as follows:

Part one: Journal additional requirements revision

Response: 

As per your advice and PLOS ONE's style requirements, the following changes are made: the first word of both the title and the subtitle, as well as all the proper nouns and generic names are modified. The postal addresses of the corresponding authors are deleted, and their initials are placed in brackets after their e-mail addresses. The fonts of the level-one headings, level-two headings and level-three headings are changed into 18pt bold, 16pt bold and 14pt bold respectively. The formats of level-four headings 3.3.2.1 Cluster head selection phase and 3.3.2.2 Networking phase are changed into (1) Cluster head selection phase and (2) Networking phase. The word "Figure" used to refer to the figures in the paper is abbreviated as "Fig". The revisions are marked in red.

2. Thank you for stating the following in the Acknowledgments/ Financial Disclosure Section of your manuscript: 

This work was supported by the National Natural Science Foundation of China (Project No. 60972127, 61072111 and 60672156) and Key Project of Jilin Provincial Science and Technology Department (Project No. 20100503) and the Project for Science and Tech-nology Center and Science and Technology Service Platform (Project No. 20180623004TC). The funders had no role in study design, data collection and analysis, decision to publish, or preparation of the manuscript.

This work was supported by the National Natural Science Foundation of China (Project No. 60972127, 61072111 and 60672156) and Key Project of Jilin Provincial Science and Technology Department (Project No. 20100503) and the Project for Science and Tech-nology Center and Science and Technology Service Platform (Project No. 20180623004TC). The funders had no role instudy design, data collection and analysis, decision to publish, or preparation of the manuscript.

Response: 

As per your advice and PLOS ONE's relevant requirements, the funding-related text is removed from our manuscript. The funding information declared in the Funding Statement section of our online submission form is correct, and "None" is presented in the Acknowledgements section.

Response: 

As per your advice, an ORCID iD has been obtained, i.e. 0000-0003-2081-9999, and it has been validated in Editorial Manager.

4. We note you have included a table to which you do not refer in the text of your manuscript. Please ensure that you refer to Table 1 in your text; if accepted, production will need this reference to link the reader to the Table.

Response: 

As per your advice, the table number "Table I" is changed to "Table 1", and a description is added to refer to Table 1 in the text of our manuscript, and the sentence "The parameter settings in the experiment are shown in Table 1." is added. The corrections are marked in red.

As per your advice, the whole reference list has been checked, and reference No.13 is changed from "Chinese Pharmacopoeia. Part 12015; 8" to "State Pharmacopoeia Commission. Chinese Pharmacopoeia. Beijing: China Medical Science and Technology Press; 2015. PP. 260-261".

Part two, Responses to the reviewer's comments:

1. Additional Editor Comments:

Further elaboration on the experiment setup is required. The rationale of the selected evaluation criteria has not be explained clearly. The connection between the results (i.e. figures) and the intrepretations in the analysis section should be highlighted and extended further.

Response:

As per your advice, further elaboration on the experiment setup is provided, Table 2, Table 3 and Table 4 are added to explain the rationale of the evaluation criteria, and algorithm comparison is highlighted and extended further with respect to the experiment results in different situations.

2. Reviewers #1:

The paper proposed an algorithm for high-efficiency wireless sensor networks for a ginseng field explained well. But parameters settings and description of the experimental parameters need more explanation. May be accepted for the publication.

Response:

As per the reviewer's advice, a detailed description of the experimental parameters is added, which is placed after Table 1 and marked in red, as follows: As shown in Table 1, 50 to 300 wireless sensors are deployed in the network at random; and their physical position cannot be changed, but such equipment has the function of transmitting power adjustment; and power consumption of the node for transmitting/receiving 1bit data Eelec=50nJ/bit.

In the proposed algorithm, Particle Swarm Optimization (PSO) is taken as the basis of algorithm optimization; and in the particle update rules, number of particles m=10, selected self-learning factors C1 and C2=0.5, inertial coefficient λ=0.5, and maximum number of optimizations of the particle p=30.

3. Reviewer #2:

The paper presents Algorithm for Wireless Sensor Networks in Ginseng Field in Precision Agriculture. The overall architecture and main component are describes. Since statistical analysis should be performed rigorously, you can add a table contains many aspects comparing existing algorithms to the proposed algorithm and specify the method of testing that had been used in the study as well.

Response: 

As per the reviewer's advice, Table 2, Table 3 and Table 4 are added, in which algorithm comparison is provided on network life cycle under different numbers of nodes, on network life cycle under difference distances of central node, and on network life cycle under different distances between central node and other nodes, as follows. In the algorithm comparison, the existing algorithms are contrasted with the proposed algorithm, and the test method is indicated. The revisions are marked in red.

Table 2 Algorithm comparison on network lifecycle under different numbers of nodes

Number of nodes

Algorithm 50 100 150 200 250 300

LEACH 471 830 1380 1890 2198 2258

BCDCP 488 960 1603 2260 2460 2680

ECHERP 493 1180 1900 2301 2597 2732

Proposed algorithm 502 1302 2301 2587 2730 2886

Table 3 Algorithm comparison on network life cycle under different distances of central node

Distance of central node

Algorithm 50 75 100 125 150 175

LEACH 1896 1800 1687 1409 1001 690

BCDCP 2386 2205 2098 1689 1201 834

ECHERP 2500 2402 2385 2045 1480 960

Proposed algorithm 2598 2523 2443 2303 1513 1000

Table 4 Algorithm comparison on network life cycle under difference distances between central node and other nodes

Number of other nodes

Algorithm 150 100 50 0

LEACH 4102 5540 7054 8579

BCDCP 6940 7602 8569 9509

ECHERP 7340 8402 9230 10301

Proposed algorithm 7580 8501 9474 10503

We tried our best to improve the manuscript and made some changes to the manuscript. These changes will not influence the content and framework of the paper. And here we did not list the changes but marked in red in the revised paper. We appreciate for Editors/Reviewers/ warm work earnestly, and hope that the correction will meet with approval.

Once again, thank you very much for your comments and suggestions.

---

## [Decision Letter · Decision Letter 1]

19 Jan 2022

Algorithm for wireless sensor networks in ginseng field in precision agriculture

PONE-D-21-28152R1

Dear Dr. Chen,

We’re pleased to inform you that your manuscript has been judged scientifically suitable for publication and will be formally accepted for publication once it meets all outstanding technical requirements.

Kind regards,

Chi-Tsun Cheng, Ph.D., M.Sc., B.Eng.

Academic Editor

PLOS ONE

Additional Editor Comments (optional):

All issues have been resolved in the current version. The paper is recommended to be accepted.

Reviewers' comments:

Reviewer's Responses to Questions

**Comments to the Author**

1. If the authors have adequately addressed your comments raised in a previous round of review and you feel that this manuscript is now acceptable for publication, you may indicate that here to bypass the “Comments to the Author” section, enter your conflict of interest statement in the “Confidential to Editor” section, and submit your "Accept" recommendation.

Reviewer #1: All comments have been addressed

Reviewer #2: All comments have been addressed

2. Is the manuscript technically sound, and do the data support the conclusions?

Reviewer #1: Yes

Reviewer #2: Yes

3. Has the statistical analysis been performed appropriately and rigorously? 

Reviewer #1: Yes

Reviewer #2: Yes

4. Have the authors made all data underlying the findings in their manuscript fully available?

Reviewer #1: No

Reviewer #2: Yes

5. Is the manuscript presented in an intelligible fashion and written in standard English?

Reviewer #1: Yes

Reviewer #2: Yes

6. Review Comments to the Author

Reviewer #1: (No Response)

Reviewer #2: (No Response)

7. PLOS authors have the option to publish the peer review history of their article (what does this mean?). If published, this will include your full peer review and any attached files.

Reviewer #1: No

Reviewer #2: No

---

## [Editor Report · Acceptance letter]

27 Jan 2022

PONE-D-21-28152R1 

Algorithm for wireless sensor networks in ginseng field in precision agriculture 

Dear Dr. Chen:

I'm pleased to inform you that your manuscript has been deemed suitable for publication in PLOS ONE. Congratulations! Your manuscript is now with our production department. 

Kind regards, 

on behalf of

Dr. Chi-Tsun Cheng 

Academic Editor

PLOS ONE